# The Phenomenon of Anti-Drug Antibodies in Psoriasis: Mechanisms, Clinical Impact, and Therapeutic Strategies

**DOI:** 10.3390/ijms26199616

**Published:** 2025-10-02

**Authors:** Anna Mennella, Loredana Frasca

**Affiliations:** National Center for Global Health, Istituto Superiore di Sanità, ISS, 00161 Rome, Italy; anna.mennella@iss.it

**Keywords:** psoriasis, biological therapy, anti-drug antibodies (ADA)

## Abstract

Biological drugs have revolutionized the treatment of many chronic diseases, starting with cancer. They normally consist of antibodies that are also effectively used to treat several autoimmune diseases, including psoriasis. These products, called biologics, work by selectively blocking the activity of certain targets, mainly cytokines, which play a crucial role in the pathogenic and inflammatory processes involved in a particular disease. Unfortunately, a reduction in response to these biological treatments may occur over time, and this phenomenon is often due to the development of antibodies against the therapeutic antibodies. The immune responses directed to these therapeutics range from transient anti-drug antibodies (ADA) formation, with no clinical significance, to the generation of high titers and persistence of ADA, causing loss of efficacy. Considering the costs associated with the use of biological drugs, there is growing interest in identifying biomarkers that can predict clinical response to personalize treatments.

## 1. Introduction

### 1.1. Definition of Anti-Drug Antibodies (ADA)

Immunogenicity reflects the ability of an antigen to elicit a humoral or/and cell-mediated immune response. Due to unwanted immunogenicity, some patients treated with therapeutic proteins, such as antibodies that block crucial disease-mediating factors, mount anti-drug antibodies (ADA) [1]. Although a lot of scientific effort has concerned antibody engineering to reduce the immunogenicity of biological drugs, ADA development remains the basis of the ineffectiveness or adverse reactions seen in a certain percentage of patients. The clinical effect of immunogenicity is variable, ranging from no measurable effect to serious adverse events such as anaphylactoid reactions [1,2]. Theoretically, ADA can affect treatment efficacy by lowering exposure to free active drug via neutralization and/or enhanced clearance. When enough free drug is available to bind to its biological target, ADA may not have a clinical consequence [2].

The presence of ADA needs to be carefully considered, since it can cause serum drug levels to drop to sub-therapeutic levels [3], or result in loss of clinical response as a consequence of drug neutralization [4]. In addition, ADA can also contribute to pathogenic reactions at the injection site and systemically, including thromboembolic events and serum sickness [5]. ADA are usually detectable 2–4 weeks following the first biological inhibitor administration. The development of these antibodies is often reported to be of the immunoglobulin G (IgG) class, but in some instances, IgG4 production occurs [3]. IgG4 ADA are generally less inflammatory, because they are unable to activate complement and have a low affinity for Fc receptors, but they may still be able to compete with the therapeutic antibodies for the binding to the target molecule, thereby neutralizing the drug and lowering the clinical response [6]. In the absence of help from T lymphocytes (T-cells), ADA can be at low titer and consist of transient IgM, with little clinical impact. IgE antibodies, generally associated with hypersensitivity responses, are detected rarely, and IgA antibodies are not expected to develop, since none of therapeutic antibody used in therapy is administrated intra-mucosally, but only subcutaneously [7].

### 1.2. Type of Blocking Therapy with Therapeutic Antibodies and Their Relevance in Psoriasis (PsO)

Psoriasis is a Th1-driven disease, due to the elevated levels of IFNγ, TNFα, IL-17, and IL-22 found in lesions, all of which are T-cell derived cytokines. Even though IFNγ can play a crucial role in skin inflammation [8], it is not the optimal therapeutic target in the treatment of chronic plaque psoriasis. The focus has therefore shifted to Th17-related cytokines (IL-17 in particular). The current view of the pathogenesis of psoriasis relies on pathogenic Th17 cells, which can also produce IL-22 and IL-21, arising from an unknown trigger in genetically predisposed individuals as result of the production of Th17 cytokines [9]. LL37, released by damaged keratinocytes in complex with nucleic acids released by dying cells (or by neutrophil extracellular trap releasing neutrophils, called NETing neutrophils, infiltrating lesional skin), is an important stimulus for both innate and adaptive immunity activation [10]. LL37, proposed as a possible autoantigen that drives Th1/Th17 cells, also drives the activation of cutaneous plasmacytoid and myeloid DCs [10]. Th17 lymphocytes recirculate in the dermis and further promote the inflammatory cascade and persistent proliferation of keratinocytes via the release of effector cytokines, most importantly IL-17A. Anti-IL17 monoclonal antibodies represent an effective therapy in the treatment of psoriasis. Following five years of phase trials, three drugs targeting IL-17 are now available and have demonstrated high efficacy and tolerability for moderate-to-severe psoriasis [11]. Interestingly, another Th17 cytokine, IL-22, which also contributes to keratinocytes hyperproliferation and with promising pre-clinical data, has never passed a phase I trial NCT00563524 (https://clinicaltrials.gov/study/NCT00563524, accessed on 22 September 2025). This may be because the pathogenic function of IL-22 shows redundancy with other members of the IL-20 subfamily, such as IL-19 and IL-20, making its blockade clinically negligible. Furthermore, after the discovery of IL-23 as an upstream cytokine in the pathogenesis of psoriasis, efforts deviated towards generation of inhibitors binding to IL-23. Since 2018, three IL-23 inhibitors (guselkumab, tildrakizumab, risankizumab) have been approved as effective treatments for psoriasis. Guselkumab is a fully human IgG1 monoclonal antibody, tildrakizumab is a humanized IgG1 monoclonal antibody, and risankizumab is a humanized IgG1 monoclonal antibody. In humanized antibodies, only the complementarity-determining regions (CDRs) of the variable (V) regions are of mouse origin [12]. All of them bind the p19 subunit of IL-23. Initial evidence for a functional role of IL-23 in psoriasis included the clinical efficacy of an anti-p40 monoclonal antibody (blocking both IL-12 and IL-23) in psoriasis [13] and the association of a single nucleotide polymorphism in the IL23R gene in psoriasis [14]. The main mechanism behind the efficacy of these inhibitors is the inhibition of IL-17 responses. Tumor necrosis factor-α (TNFα) was originally identified as a serum factor that can induce hemorrhagic necrosis of tumors. Due to its induction of an important systemic inflammation, TNFα could never be efficiently utilized for tumor therapy [15]. Dysregulation or unimpaired TNFα production can lead to autoimmune diseases such as rheumatoid arthritis (RA), Crohn’s disease, ulcerative colitis (UC), or psoriasis, making TNFα a logical and promising therapeutic target. In chronic phase of plaque psoriasis, there are multiple sources of TNFα, and several cell types can produce it. In the skin, at the site of inflammation, keratinocytes [16], dendritic cells [17], macrophages [18], T-cells [19], and adipocytes [20] are all thought to be major producers of TNFα. Currently, four TNFα inhibitors are approved for the treatment of psoriasis: infliximab, adalimumab, etanercept, and certolizumab pegol. Infliximab is a human-murine chimeric monoclonal IgG1 antibody with a murine variable region and a human constant domain, that functions by binding TNFα ligand and thereby preventing interaction with TNFα receptors. Adalimumab is a fully human antibody that works in the same manner as infliximab. Etanercept is an engineered compound composed of TNFR2 fused to a human IgG1 Fc domain and acts by competitively inhibiting the binding of TNFα to its receptors. Certolizumab pegol is a Fab fragment of a humanized monoclonal antibody and is PEGylated to increase its stability and half-life and to decrease its immunogenicity [21]. A fifth TNFα inhibitor, golimumab, a fully human antibody, is currently approved for the treatment of psoriatic arthritis (Table 1).

## 2. Mechanisms of ADA Formation

A search was conducted in the scientific literature (PRISMA protocol not followed), on PubMed and Google Scholar, using the following keywords: “Immunogenicity of biologics in psoriasis”, “anti-drug antibodies development in psoriasis”, “ADA in psoriasis patients under biological drugs”, “Immunogenicity of mAbs used in psoriasis”, and “Side effects of biological therapy in psoriasis”. No time limit was placed on the research performed. We included clinical trials, review articles, epidemiological studies, and in vitro studies.

### Factors Influencing ADA Development: Drug-Related Factors, Patient-Related Factors, and Treatment-Related Factors

Multiple factors can contribute to the development of an immune response to the monoclonal antibodies used in therapy. These are classified in “drug-related factors” (for example) presence of T-cell and B-cell epitopes within the amino acidic sequence of the molecule), “patient-related factors”, and “treatment-related factors” (Figure 1). Drug-related factors include antibody sequence and structure, chemical modifications (for example PEGylation or fusion proteins), degree of “foreignness” (species origin, nature of the therapeutic antibody), presence of impurities and/or contaminants (aggregates, degradation products), and formulation. Patient-related factors include genetic background and disease state, especially MHC polymorphisms. There is evidence showing that the concomitant use of certain disease-modifying anti-rheumatic drugs (DMARDs) help to maintain efficacy of the therapeutic antibodies, prolonging drug availability and resistance by reducing the immunogenicity. For example, the use of methotrexate in biological therapies may attenuate the incidence of ADA in several diseases (likely due to immune suppression). The concentration of the target biological factor can influence its immunogenicity. For instance, it has been shown that patients with lower levels of TNFα prior to adalimumab treatment had a higher frequency of ADA against adalimumab than patients with higher amounts of TNFα [6]. Furthermore, treatment-related factors include dose level, treatment duration and schedule (frequency of doses), and administration route. ADA development has been reported as an important barrier for dermatologists to apply biologics dose reduction in the course of psoriasis treatment [22]. In 2022, Atalay and colleagues showed that in psoriasis patients with stable low disease activity on etanercept, adalimumab, and ustekinumab treatment, a reduction in dose was feasible, because drug concentration did not increase immunogenicity [23].

## 3. Prevalence of ADAs in Psoriasis

### 3.1. Summary of Reported ADA Rates for Different Biologics

Despite the use of comparable assays and similar time points for sample collection, ADA incidences in response to different biologics cannot be directly compared. However, the differences in incidence are also reflected in the reported proportions of patients in whom ADA impacts drug levels and clinical outcomes. This indicates that true differences in the immunogenicity of biologics used in psoriasis exist. The immunogenicity of a given drug has proven difficult to predict, as it is determined by several factors, such as the presence of T-cell and B-cell epitopes, which combine with the patient’s genetic background, as well as by the target interaction/avidity together with the route of administration.

#### 3.1.1. TNFα Inhibitors (Adalimumab, Infliximab, Certolizumab, and Etanercept)

ADA to TNFα-blocking antibodies appear almost exclusively restricted to specific drug epitopes (idiotypes), and recently, different anti-TNFα specific epitopes have been identified [7]. Antibody responses to anti-TNFα and to the majority of monoclonal antibodies used in therapy appear highly restricted to the antigen binding site, so they are predominantly neutralizing. All TNFα inhibitors on the market can be potentially immunogenic, with the incidence of ADA induction being different among them. It is reported that up to 44% of patients treated with infliximab exhibit ADA development after six months of treatment [24]; 19% of patients treated with adalimumab exhibit ADA development within the first six months of treatment, and this increase to 28% within three years [6]. For certolizumab, 37–65% of the patients treated show ADA during the therapy [25]; the percentage of patients developing ADA during etanercept therapy is around 6% [26].

The development of ADA has also been detected in response to other biologics approved for many chronic diseases, such as rituximab, abatacept, certolizumab, etanercept, secukinumab, tocilizumab, and ustekinumab. The antigenic site is largely related to the antibody-binding site; however, ADA targeting the antibody hinge regions of abatacept and etanercept have also been described [2].

#### 3.1.2. IL-12/23 Inhibitor (Ustekinumab)

Antibodies against ustekinumab have predominantly neutralizing properties, but their prevalence is relatively low. In 2023, Poor and colleagues published that anti-drug antibody formation against the anti-IL12/23 ustekinumab was not associated with impairments in drug efficacy, safety, or trough levels. They also demonstrated how the concomitant administration of methotrexate (MTX) had no significant impact on ADA formation [27]. ADA against ustekinumab could be associated with modified drug clearance through the formation of immune complexes, consequently impairing clinical response.

#### 3.1.3. IL-17 Inhibitors (Secukinumab, Brodalumab, and Ixekizumab)

Secukinumab and brodalumab are both fully human antibodies, which renders them less immunogenic compared to murine or humanized antibodies [12,28,29]. Ixekizumab is a humanized antibody that is potentially more immunogenic. The incidence of ADA to secukinumab is between 0 and 5.5%, and when present, it does not change the pharmacokinetic profiles, even when the induced antibodies have neutralizing properties [30,31]. Regarding brodalumab, the incidence of ADA is between 0 and 3.3%, the majority of ADA in response to brodalumab are transient and do not influence drug levels [32]. Reich and colleagues in 2018 reported an incidence of ADA to ixekizumab between 11 and 19.4% [33]. Like the other IL-17 inhibitors, ADA to ixekizumab does not change pharmacokinetic profiles.

#### 3.1.4. IL-23 Inhibitors (Guselkumab, Risankizumab, and Tildrakizumab)

The incidence of ADA against IL-23 blocking antibodies is reported as follow: 4.1–14.7% with guselkumab, 6.51–18% with tildrakizumab, and 14.1–31% with risankizumab [34]. For all IL-23 inhibitors, ADA positivity did not appear associated with a change in drug efficacy, even though in 2019 it was shown that psoriasis patients positive for neutralizing ADA against tildrakizumab experienced lower serum levels and reduced efficacy of the drug [35]. In 2023, Armstrong and colleagues showed that there was no impact of ADA on guselkumab pharmacokinetics, since drug concentrations were comparable between ADA-negative and ADA-positive patients.

Table 2 summarizes the most relevant clinical trials that evaluate the prevalence of ADAs in psoriasis patients under different biological therapies.

## 4. Clinical Consequences of ADA Formation

### 4.1. Reduced Drug Efficacy and Loss of Response, Drug Neutralization by Anti-Drug Antibodies: Comparison Between Neutralizing Versus Non-Neutralizing ADA

ADA that develop in patients treated with a biological drug belong to two main categories: non-neutralizing antibodies (non-nt-ADA) and neutralizing antibodies (nt-ADA). The first class of antibodies, also known as binding antibodies, specifically bind the biological drug, but do not interfere with the interaction between the drug and the target. The second type (nt-ADA) binds directly to, or in close proximity to, the pharmacologically active site of the biological drug, physically affecting the ability of the drug to bind to the target [44]. However, both neutralizing and non-neutralizing antibodies are clinically relevant. Nt-ADA directly act by reducing the drug efficacy, while non-nt-ADA may indirectly act on therapeutic efficacy by lowering the drug systemic exposure, which can result from accelerated drug clearance, resulting in a clinically similar outcome. In addition, both types of ADA can form immune complexes after binding to the drug, which are easily removed within days [3]. However, the presence of nt-ADA and non-nt-ADA does not simply correlate with reduced therapeutic effect, as this effect depends on the balance between drug levels and ADA levels [45].

### 4.2. Increased Drug Clearance and Altered Pharmacokinetics

Generally, all the therapeutic mAbs belong to IgG class and are administered either subcutaneously or intravenously. It is believed that their elimination is provided by non-specific pinocytosis and endocytosis, and by target-mediated drug disposition (TMDD) mechanisms leading to proteolysis [46]. Theoretically, the clearance of mAbs is faster when higher amounts of the target are present. This is not the case for anti-TNFα/TNFα, since the serum concentration of anti-TNFα is always higher than that of TNFα. Consequently, anti-TNFα/TNFα binding does not normally contribute to the elimination of the inhibitor [47,48,49]. On the other hand, ADA/anti-TNFα binding can greatly increase elimination rates of the drug. It is estimated that, in the presence of detectable ADA, the clearance of adalimumab increases by 4-to-5.5-fold [40,41]. Recently, different assays have become available to measure neutralizing antibodies, such as cell-based in vitro assays and competitive ligand-binding assays. The main issue with the use of these tests is that it is impossible to mimic the exact in vivo neutralization in a single in vitro assay, where the concentrations of the component exceed the variation seen in patients. Therefore, nt-ADA positivity does not necessarily indicate that neutralization is happening in vivo, and nt-ADA negativity indicates only that neutralization is not detectable, but does not exclude that it can still happen in vivo [45]. Moreover, crystallography experiments indicate that ADA behave as non-nt-ADA in vitro could still act as neutralizing antibodies. Indeed, they can bind the biological drug at a site different from the active site, but they physically occupy the space meant for the drug target [50]. Consequently, most ADA have an innate or structural capacity to neutralize the biologic therapeutic.

## 5. Strategies to Minimize ADA Development

### 5.1. Combination Therapy with Immunosuppressants (Such as Methotrexate)

Individual variation in susceptibility to ADA development is also related to other medications that a patient is taking [51]. As mentioned above, ADA formation may be prevented by co-administration of methotrexate (MTX), which has been well investigated in patients with psoriasis receiving anti-TNFα treatment. The addition of MTX, even after the development of ADA, potentially provides an alternative treatment strategy in patients who develop secondary loss of efficacy of the drug [52,53]. MTX is able to decrease the clearance therapeutic mAbs and reduce its immunogenicity. It has been shown that psoriasis patients receive MTX subcutaneously starting with 7.5–10 mg, reaching 12.5–20 mg/week [54,55]. Azathioprine is an immunosuppressive drug with an effect similar to that of MTX. It has been observed to reduce immunogenicity and ADA formation, in combination with infliximab or adalimumab [56]. Concomitant therapy with mycophenolate and leflunomide (100 mg daily for the first days, then 20 mg for three months [57]) has also been shown to associate with lower ADA prevalence [58]. In addition, there are non-specific methods that generally suppress the immune system are used as a pre-treatment strategies, such as anti-CD20, anti-CD25, and anti-CD22 mAbs, which result in B-cell depletion [59]. In 2018, Sauna et al. proposed the use of bortezomib, a proteosome inhibitor, to sustain immune tolerance to mAbs [60].

### 5.2. Drug Dose Optimization and Interval Adjustments

It has been shown that a high dose of therapeutic mAbs induces a state of immune tolerance to biological treatment, with less frequent of ADA generation [56]. However, large doses of therapeutics are extremely expensive and may cause additional drug-related side effects. Publications on biological drug reduction (anti-TNF-α, anti-IL-12/23, anti-IL-17, and anti-IL-23) in psoriasis have shown good clinical effectiveness and safety, as well as cost savings.

## 6. New Approaches in the Field for Psoriasis and Other Diseases

Many scientific studies are now focusing on new possible targets for the treatment of autoimmune diseases, including psoriasis. Anti-interferons (anti-IFNs) have been tested in autoimmune diseases such as lupus erythematous (LE), but they have not proven efficient in chronic plaque psoriasis due to the presence of two inflammatory pathways in psoriasis [61]. Other approaches under examination include targeting plasmacytoid dendritic cells (pDC) with anti-ILT-7 or anti-BDCA2, or with Toll-like receptor inhibitors (for TLR7 and TLR9), which are responsible of pDC activation and type I IFN production in psoriasis. Although biologics are more efficacious and safer than non-specific immunomodulators, new strategies are under study to reduce unwanted effects and maintain long-term efficacy. New compounds targeting TNFR1 now exist (ATROSAB, MDS5541, and TROS) [62] and are being tested for effectiveness. Recently, anti-TNFα biosimilars have been produced at reduced costs. An infliximab biosimilar has been tested in chronic plaque psoriasis with success and is now approved [63]. Other approaches include dual cytokine inhibition, or even biologics with dual-specificity (bi-specific antibodies). Although concurrent TNFα and IL-1 inhibition has not yielded increased efficacy for the treatment of RA [64], novel approaches are being undertaken, such as the use of bi-specific antibodies targeting IL-17 and TNFα [65]. Due to the synergistic effect of TNFα and IL-17 in inflammation, this may increase efficacy for patients with limited therapeutic responses to single cytokine blocking. More recently, the technology for generating antibodies carrying three specificities (tri-specific antibodies) has been developed [66]. In psoriasis this could mean that specificity number 1 will target tissue-specific antigens in the skin, specificity number 2 will block cytokines, and specificity number 3 will deliver anti-inflammatory compounds to the targeted site [67].

## 7. Clinical Management of Patients with ADA

### 7.1. ADA Monitoring and Testing Methods

Detection and analysis tests for ADA formation are helpful tools in understanding potential immune responses in patients undergoing biological therapy. The scientific data on ADA and their impact on serum drug concentrations and clinical outcomes are generated by using different assay methods, which makes it difficult to compare the results across various studies [68]. The first studies on the immunogenicity of monoclonal antibodies for therapy used “drug-sensitive assays”, but currently, the majority of protocols operate with “drug-tolerant assays”. The first type of test often leads to drug interference, which means an underestimation of immunogenic potential. This is happening because when a biologic is present in the serum, it can form complexes with ADA, and these complexes mask binding to the detection reagent. To overcome these effects, researchers have developed, over the years, several “drug-tolerant assays” to detect and measure ADA bound to the therapeutic antibody. Most of these assays use an acid dissociation step to dissociate complexes, resulting in free biologic drug and free ADA. The subsequent step usually involves either capture/removal of excess drug [69,70] or adding labeled biologic, which competes with the biologic in the sample [71,72,73]. Upon removal of the drug, detection of the free ADA is feasible with a drug-sensitive assay. Even with these drug-tolerant assays, ADA may be underestimated or missed because of the rapid elimination of ADA–drug complexes. Though small complexes may circulate for prolonged times [3], larger complexes will generally be rapidly removed. This results in the removal of both drug and ADA. Even during early ADA formation, with high biological drug levels, it is possible that a significant portion of the ADA is rapidly removed from the circulation after binding the biological drug. The tests mainly used for ADA detection include bridging ELISA, affinity capture elution ELISA (ACE), electrochemiluminescence (ECL) bridging assay, biotin-drug extraction with acid dissociation (BEAD) assay, radioimmunoassay, temperature-shift radioimmunoassay, and homogeneous mobility shift assay (HMSA) [45]. None of the currently available assays can detect all different forms of antibodies: ELISA may favor the detection of IgM antibodies, whereas radioimmunoassay can detect IgG4 antibodies, which are reported to have a greater potential for neutralization [74]. Other methods that have also been tested for the analysis of ADA include the use of capillary electrophoresis (CE), reporter gene assays (RGA), surface plasmon resonance (SPR), or LC–MS (liquid chromatography–mass spectrometry). Anyway, it is necessary to develop greater standardization of ADA detection methods.

### 7.2. Switching Strategies: Intra-Class Versus Inter-Class Switching

Clinical guidelines suggest changing biologic classes after primary treatment failure and considering switching within the same class or to a different class for secondary failures [75]. Biological therapy switching is necessary in the management of moderate-to-severe psoriasis. When patients under mAbs start to show adverse events against medication, with the presence of anti-drug antibodies leading to efficacy loss, clinicians decide between intra-class and inter-class switching.

Intra-class switching refers to changing biologics within the same class, such as switching from one anti-IL-17 to another one (e.g., secukinumab to brodalumab) or from one anti-IL-23, IL-12/23, or TNF-α inhibitors to another targeting the same cytokine. By contrast, the switching that involves shifting to a biologic from a distinct class, is known as inter-class switching, e.g., moving from an IL-17A inhibitor to an IL-23 inhibitor, or vice versa. In 2025, Zhang and colleagues published a meta-analysis showing that both intra-class and inter-class switching are effective when monitoring clinical outcomes. In particular, intra-class switching has a superior efficacy in the short term, and both switching strategies share same efficacy in the long term [76]. There are many publications on intra-class switching for psoriasis patients, which are good therapeutic options for TNFα, IL-17, IL-23, and IL-12/23 inhibitors [76,77,78,79]. In the case of ADA development, intra-class switching could be a good strategy since ADA are specific to a single biologic, and there is no cross-reaction within the same group [80]. Therefore, if different biologic agents within the same group fail, another biologic from a new group could still work. Inter-class switching sometimes requires more time to show its effectiveness because it targets different pathways and, to elicit an optimal response, serum drug levels need to reach a peak. Ruiz-Villaverde and colleagues in 2022 showed that guselkumab may be an effective alternative for those intolerant or unresponsive to other biologics [81]. In addition, ustekinumab is shown to be an effective therapeutic option after failure of other TNFα blocking agents [82].

### 7.3. Emerging Techniques and Biomarkers for Predicting ADA Risk

The search for predictors of a favorable clinical response to biologics in patients with inflammatory diseases, such as psoriasis, has been the focus of several publications [83,84,85,86]. In 2024, a study from Kraev and colleagues constructed multiple mathematical models with disease activity index, serological, immunological, and demographic data aimed at predicting the development of neutralizing ADA to anti-TNFα. They have found that negative serological status, alcohol abstinence, lower body weight, younger age, and smoking cessation could be used as reliable predictors of anti-TNFα treatment efficacy [87]. With advances in laboratory technology and computing, it is now feasible to put together the results coming from in vitro assays with in silico prediction tools at the beginning of the drug development to produce less immunogenic products. One of the first in silico prediction tools was the T-cell epitope predicting algorithms (TCEs), based on antibody sequence, which should be confirmed in vitro [43,76]. The newest tools such as NetMHCIIpan [88], use data on peptide-MHC binding affinity obtained by mass spectrometry. In addition, B-cell epitope predicting algorithms (BCE) use the physical properties of amino acids or known antibody-binding regions to predict BCEs in therapeutic mAbs [89]. The union of TCE and BCE has generated the Immune Epitope Database (IEDB), which uses a large library of pre-established B- and T-cell epitopes and is one of the most widely used in silico prediction database. Other known tools such as SWISS-MODEL can predict the 3D structures of mAbs, and programs such as AlphaFold can predict the secondary structure of mAbs de novo [51]. Machine learning methods have been used only occasionally to predict the risk of non-response to biological drugs. In 2024, Ukalovic and colleagues, in the context of RA, developed a prediction model for five different biologics using machine learning methods based on patient data derived from a registry (named BioReg), in order to identify patients with a high risk of non-response before therapy [90]. Machine learning and artificial intelligence tools may be beneficial in predicting the clinical response or risk of toxicity to biological therapies. By classifying diverse drug-related factors and analyzing their characteristics together with predictive models, machine learning could be crucial in drug development and in stratifying patients before therapy. Altogether, in vitro data, in silico studies, machine learning, and artificial intelligence tools, can be useful in therapeutic mAbs development phases, helping to increase the accuracy of biological drugs and their clinical responses (Figure 2).

## 8. Conclusions

The problem of development of ADA during biological therapy is challenging, and methods to detect ADA are an important area of research in clinical settings. Clinicians should be aware of the development of ADA and consider testing whether this type of reactivity develops in their patients. Given the costs associated with the use of biological drugs, there is growing interest in identifying biomarkers that can predict clinical response, thus personalizing treatments. Currently, there are no biomarkers that can be used effectively in routine clinical practice [91].

Moreover, studies on ADA have revealed that some of the drugs used are less or more immunogenic than others. This can also reveal a hierarchy in the choice of the type of drugs to be used after the failure of the first treatment.

Therefore, it is really crucial to report information on immune responses observed during clinical trials, as well as in post-marketing observational studies, to be aware of the incidence of ADA development with each drug. Any implications of ADA responses on pharmacokinetics, pharmacodynamics, safety, or efficacy, for any therapeutic mAbs development program should be carefully taken into consideration for better control of chronic diseases with biological therapies.

## Figures and Tables

**Figure 1 ijms-26-09616-f001:**
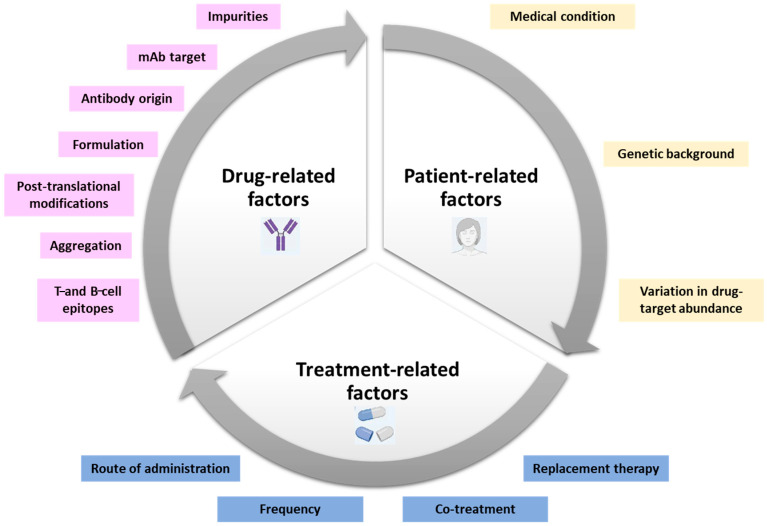
**Factor influencing the efficacy of therapeutic antibodies in autoimmune diseases**. Three main categories of factors influence biological therapy in autoimmunity, with a comparable contribution among patient backgrounds, formulation, and therapeutic protocols.

**Figure 2 ijms-26-09616-f002:**
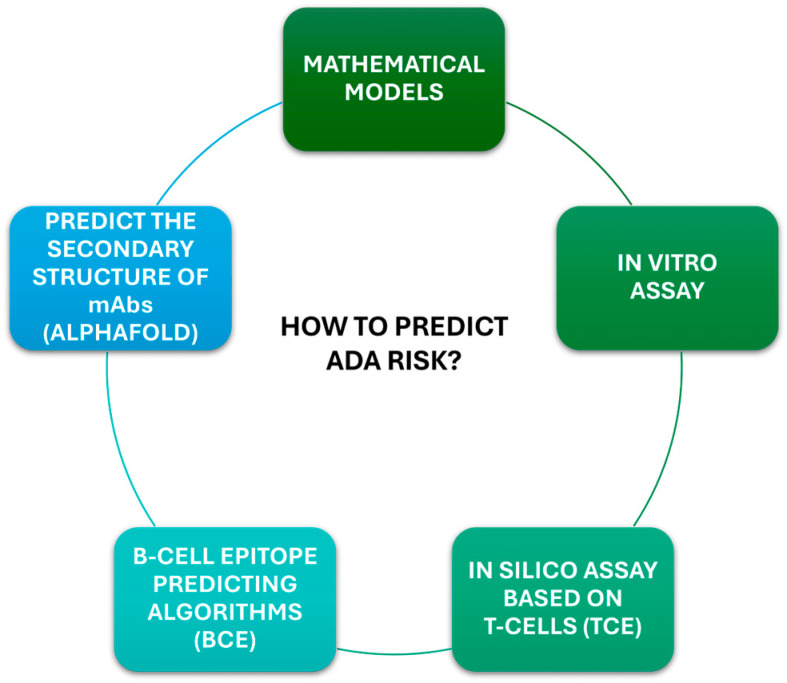
Techniques and biomarkers for predicting ADA risk. Five main tools are used to predict anti-drug antibodies (ADA) formation risk in different phases of drug development.

**Table 1 ijms-26-09616-t001:** Classification of monoclonal antibodies used in psoriasis therapy.

Class/Target.	Drugs (mAbs)	Mechanism of Action	Clinical Notes
anti-TNFα	Infliximab, Adalimumab, Golimumab, Certolizumab pegol	Neutralization of TNFα: reduced inflammation	First to be introduced; effective but with risk of opportunistic infections
anti-IL-12/23 (p40)	Ustekinumab	Blocks p40 subunit shared by IL-12 and IL-23	Acts on both Th1 and Th17 pathways
anti-IL-23 (p19 selective)	Guselkumab, Risankizumab, Tildrakizumab	Selective blockade of IL-23 p19 subunit	High specificity and excellent efficacy and safety
anti-IL-17A	Secukinumab, Ixekizumab	Neutralize IL-17A	Rapid clinical efficacy, risk of mucocutaneous candidiasis
anti-IL-17RA	Brodalumab	Blocks IL-17 receptor A (IL-17RA): inhibits multiple isoforms	Very potent but with warning for suicidal ideation risk
Others in development/less widely used	Sonelokimab (nanobody anti-IL-17A/F), Bimekizumab (anti-IL-17A/F)	Block multiple IL-17 isoforms	In late-stage development or already approved in some countries

**Table 2 ijms-26-09616-t002:** Relevant clinical trials describing the percentage of ADA development in psoriasis patients under treatment with biologics.

Biologics	Key Trials/Program	Phase Population	Reported Treatment-Emergent ADA	Reported Clinical Impact of ADA	Source (Main)
Infliximab	Pivotal psoriasis trials and multiple observational cohorts	Phase 3_moderate-to-severe plaque psoriasis	High variability. Clinically meaningful ADA rates were reported, with antibodies reported in many patients in observational series (rates higher than fully human mAbs).	ADAs are frequently associated with lower serum drug levels and an increased risk of loss of response and infusion reactions. Concomitant immunosuppression reduces ADA development.	Anjie et al. [36]
Adalimumab	Pivotal psoriasis trials and biosimilar comparators	Phase 3_large psoriasis populations	Variable (5–20% reported in different studies and assays); biosimilar studies report similar ADA levels.	ADAs (especially neutralizing) can reduce trough levels and are related to diminished efficacy in some patients. Assay/definition dependent.	Valenzuela et al. [37]
Etanercept	Pivotal trials (e.g., etanercept registration trials) and long-term cohort studies	Phase 3_moderate-to-severe psoriasis	Low to modest ADA detection. Many assays find non-neutralizing antibodies or low clinical relevance.	When present, ADAs are often non-neutralizing and less linked to loss of response than with monoclonal TNF agents.	Hu et al. [38]
Ustekinumab	PHOENIX/ACCEPT biosimilar trials and pooled analyses	Phase 3_moderate-to-severe psoriasis	Low (typically single digit %; assay dependent).	Most studies report low ADA rates. When ADAs occur, they can be associated with lower drug levels, but clinical impact is often limited and assay dependent. Improvements in assay methods have changed reported rates over time.	Loeff et al. [39]
Secukinumab	ERASURE/FIXTURE pooled safety programs	Phase 3_moderate-to-severe psoriasis	Very low (~0.4%) ADA reported in pooled Phase 3 analyses.	ADA incidence is very low. Pooled analyses report no clear association with loss of efficacy or altered pharmacokinetics.	Karle et al. [40]
Ixekizumab	UNCOVER-1, UNCOVER-2, and UNCOVER-3 (phase 3 program)	Phase 3_moderate-to-severe psoriasis	Low-to-moderate. Neutralizing ADAs reported in pooled analyses. 5–8% had neutralizing ADA.	High-titer neutralizing ADAs are linked to reduced serum concentrations and, in some cases, loss of efficacy. Most patients remain ADA-negative, and efficacy is maintained.	Gordon et al. [41]
Brodalumab	AMAGINE-1/AMAGINE-2/AMAGINE-3 Phase 3 program and pooled analyses	Phase 3_moderate-to-severe psoriasis	Very low ADA rates were reported in trials.	Low immunogenicity. When ADAs appear, they have not been a major driver of loss of efficacy in pooled studies data.	Farahnik et al. [42]
Guselkumab	VOYAGE-1/VOYAGE-2, and other phase 3 trials	Phase 3_moderate-to-severe psoriasis	Low ADA rates were reported in VOYAGE pooled data.	ADAs are reported but generally not clinically meaningful (no consistent effect on efficacy or pharmacokinetics).	Zhu et al. [43]

## Data Availability

No new data were created or analyzed in this study. Data sharing is not applicable to this article.

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
