# Peer review of "The Phenomenon of Anti-Drug Antibodies in Psoriasis: Mechanisms, Clinical Impact, and Therapeutic Strategies"

_ijms, 2025, doi:10.3390/ijms26199616_

Round 1
Reviewer 1 Report
Comments and Suggestions for Authors
The authors present a very thorough, excellent review of anti-drug antibodies in patients with psoriasis. This is of much interest, since we experience this problem every day in clinical practice, and clinicians need to be aware of ADAs as well as of strategies to mitigate them. Maybe the authors could just add a sentence with doses of methotrexate or leflunamide that are commonly used, since they are lower than those for the treatment of psoriasis.
A "weakness", which does not need to be addressed, is the fact that measuring ADA's is expensive, and rarely performed in the clinic, since, except on rare occasions, it is not relevant to the choice of another biologic.
Author Response
We thank that reviewer for the positive evaluation of our review.

Reviewer 2 Report
Comments and Suggestions for Authors
We have read with interest the review presented by Mennella & Frasca, who compiled information on the anti-drug antibodies in psoriasis: mechanisms, clinical impact, and therapeutic strategy. The authors clearly and concisely explain the problem of anti-drug antibodies (ADA), their formation mechanisms, their clinical consequences and strategies to counteract their development and the management of patients with ADA. However, to ensure the clarity and soundness of the findings presented, several points must be clarified before the work can be accepted for publication in the International Journal of Molecular Sciences.
- The authors ought to specify the databases that were reviewed, the duration of the analysis, the total number of studies examined, and the keywords utilized, to provide context for the search.
- Furthermore, it is recommended that the authors outline the inclusion criteria for the selection of the articles reviewed.
- The authors should incorporate a table that summarizes the clinical trials reviewed to evaluate the prevalence of ADAs. This table should detail the biologic medications administered, the type of psoriasis addressed, the design of the clinical trial, the number of participants, the duration of treatment, and the prevalence of ADAs.
- The authors need to more accurately define the concepts of intra-class and inter-class switching.
- The authors mention that machine learning and artificial intelligence tools may be beneficial in predicting the clinical response to biological therapies. Given the significance of this for future developments, it would be advantageous to reference some recent studies on this subject.
- Considering the interest surrounding this topic, we propose that the authors include a table or figure that outlines the most pertinent biomarkers utilized to predict the risk of developing ADAs, including their advantages and limitations.
- Please revisit the bibliographic citations found in lines 164, 173, 291, and 299, as some of these citations are missing the publication year.
- The conclusion currently offers only general recommendations on the subject. The authors should revise the conclusions to present their insights on the most significant advancements related to the clinical impact and mechanisms of ADAs.
Round 2
Reviewer 2 Report
Comments and Suggestions for Authors
The authors have improved the manuscript and followed the recommendations made during the previous revision. I suggest that the article be accepted in its present form.